# Not All Platelets Are Created Equal: A Review on Platelet Aging and Functional Quality in Regenerative Medicine

**DOI:** 10.3390/cells14151206

**Published:** 2025-08-06

**Authors:** Fábio Ramos Costa, Joseph Purita, Rubens Martins, Bruno Costa, Lucas Villasboas de Oliveira, Stephany Cares Huber, Gabriel Silva Santos, Luyddy Pires, Gabriel Azzini, André Kruel, José Fábio Lana

**Affiliations:** 1Department of Orthopedics, FC Sports Traumatology, Salvador 40296-210, BA, Brazil; 2PUR-FORM, Boca Raton, FL 33432, USA; jpurita@aol.com; 3Medical School, Tiradentes University Center, Maceió 57038-000, AL, Brazil; rubensdeandrade@hotmail.com; 4Medical School, Zarns College, Salvador 41720-200, BA, Brazil; fabiocosta7113@gmail.com; 5Department of Internal Medicine, Hospital Geral Roberto Santos, Salvador 40301-110, BA, Brazil; oliveiralucasv@gmail.com; 6Department of Orthopedics, Brazilian Institute of Regenerative Medicine (BIRM), Indaiatuba 13334-170, SP, Brazil; stephany_huber@yahoo.com.br (S.C.H.); luyddypires@gmail.com (L.P.); drgabriel.azzini@gmail.com (G.A.); kruel.andre@gmail.com (A.K.); josefabiolana@gmail.com (J.F.L.); 7Regenerative Medicine, Orthoregen International Course, Indaiatuba 13334-170, SP, Brazil; 8Medical School, Max Planck University Center (UniMAX), Indaiatuba 13343-060, SP, Brazil; 9Clinical Research, Anna Vitória Lana Institute (IAVL), Indaiatuba 13334-170, SP, Brazil; 10Medical School, Jaguariúna University Center (UniFAJ), Jaguariúna 13911-094, SP, Brazil

**Keywords:** platelet senescence, platelet-rich plasma, regenerative medicine, oxidative stress, platelet biomarkers

## Abstract

Platelet-rich plasma (PRP) is widely used in regenerative medicine, yet clinical outcomes remain inconsistent. While traditional strategies have focused on platelet concentration and activation methods, emerging evidence suggests that the biological age of platelets, especially platelet senescence, may be a critical but overlooked factor influencing therapeutic efficacy. Senescent platelets display reduced granule content, impaired responsiveness, and heightened pro-inflammatory behavior, all of which can compromise tissue repair and regeneration. This review explores the mechanisms underlying platelet aging, including oxidative stress, mitochondrial dysfunction, and systemic inflammation, and examines how these factors influence PRP performance across diverse clinical contexts. We discuss the functional consequences of platelet senescence, the impact of comorbidities and aging on PRP quality, and current tools to assess platelet functionality, such as HLA-I–based flow cytometry. In addition, we present strategies for pre-procedural optimization, advanced processing techniques, and adjunctive therapies aimed at enhancing platelet quality. Finally, we challenge the prevailing emphasis on high-volume blood collection, highlighting the limitations of quantity-focused protocols and advocating for a shift toward biologically precise, function-driven regenerative interventions. Recognizing and addressing platelet senescence is a key step toward unlocking the full therapeutic potential of PRP-based interventions.

## 1. Introduction

Platelet-rich plasma (PRP) has emerged as one of the most widely adopted autologous orthobiologic strategies in regenerative medicine, particularly in musculoskeletal disorders [1]. By concentrating platelets from a patient’s own blood, PRP delivers a reservoir of growth factors, cytokines, and extracellular vesicles directly to damaged tissues, aiming to stimulate repair and modulate inflammation [2]. For over two decades, efforts to optimize PRP formulations have focused heavily on platelet concentration, activation methods and white blood cell content [3]. However, despite this progress, clinical outcomes remain inconsistent and often unpredictable. This discrepancy continues to frustrate practitioners and limit broader clinical acceptance [4].

A key factor which is often overlooked may help explain this variability: the biological age and functional quality of the platelets themselves. Recent evidence suggests that platelet senescence, the natural aging process that platelets undergo during their short circulation lifespan, may significantly impair their regenerative potential, independent of total platelet count [5]. In this light, two different PRP samples with identical platelet concentrations could yield radically different therapeutic results depending on the proportion of young versus aged platelets present.

Senescent platelets exhibit marked alterations in morphology as well as transcriptomic and proteomic profiles, reduced granule content, impaired responsiveness to activation stimuli, and even pro-inflammatory behavior [5,6]. These changes, often compounded by patient-related factors such as age, comorbidities, and oxidative stress, may compromise the expected efficacy of PRP treatments [7,8]. As a result, a growing body of research now advocates for a paradigm shift in PRP therapy, moving beyond the fixation on quantity and toward a more refined understanding of platelet quality.

This review explores the emerging concept of platelet senescence as a determinant of PRP efficacy. We examine the molecular hallmarks of aging platelets, the mechanisms driving their functional decline, and the clinical implications of their presence in orthobiologic formulations. Additionally, we discuss current and potential strategies for assessing platelet age, optimizing patient preparation, and improving PRP processing techniques to selectively harness the most therapeutically potent platelet subpopulations. By addressing this overlooked dimension of PRP biology, we aim to provide a foundation for more consistent, personalized, and effective regenerative therapies.

## 2. The Biology of Platelet Aging

Platelets are anucleate cytoplasmic fragments derived from megakaryocytes in the bone marrow, with a well-established lifespan of approximately 7 to 10 days in circulation [9]. During this limited timeframe, they progressively undergo biochemical, structural, and functional changes that define their trajectory from young, highly active cells to senescent, functionally compromised ones [5,10,11]. These gradual alterations are not merely the reflection of a passive timeline, but an active transformation shaped by environmental exposure, oxidative stress, and cumulative molecular degradation [11]. The sequential stages of platelet maturation, circulation, senescence, and reticulo-endothelial clearance are illustrated in Figure 1.

Young platelets are characterized by greater size, higher RNA content, and a more robust reservoir of functional receptors, cytoskeletal proteins, and α-granules [12]. These attributes translate into heightened reactivity, stronger aggregation responses, and a more efficient release of biomolecules [12,13,14]. In contrast, senescent platelets tend to display a distinct phenotype marked by reduced volume, lower RNA content, altered granule architecture, and loss of surface receptor expression [12,13,14]. These structural and molecular alterations impair the ability of platelets to respond to activating stimuli, thereby diminishing their capacity to contribute effectively to tissue repair [12,15,16].

Recent transcriptomic and proteomic studies have deepened our understanding of this process. For instance, comparative analyses have revealed a substantial downregulation of functional transcripts in aged platelets, alongside the acquisition of circulating extracellular RNA and proteins, which reflects a disorganized and deteriorating cellular state [17]. Rather than maintaining their intrinsic regenerative toolkit, senescent platelets often accumulate markers of oxidative damage and membrane instability [18,19], leading to premature degranulation [12,17] and therefore, a loss of therapeutic payload before reaching the target tissue.

The biological triggers of platelet aging include both intrinsic factors, such as the progressive depletion of mitochondrial reserves and antioxidant capacity, and extrinsic stimuli, including exposure to reactive oxygen species (ROS) and systemic inflammatory mediators [16]. These forces collectively promote a functional decline that extends beyond hemostasis, affecting the platelet’s ability to orchestrate cellular signaling, immunomodulation, and angiogenesis [16].

Importantly, the clinical relevance of these changes becomes apparent when comparing the biological performance of PRP samples enriched with different proportions of young and senescent platelets. The functional heterogeneity within a single PRP preparation, often overlooked in current protocols, may be a major determinant of clinical outcomes. Recognizing platelet age as a biological variable introduces a new layer of complexity to PRP-based therapies and demands a shift in both laboratory processing and patient selection strategies. While platelet quality (including mitochondrial health, receptor integrity, and granule content) strongly influences responsiveness to activation stimuli, it is not the only determinant. Variables such as the type and strength of the activation method, the time elapsed from blood draw to use, and interactions with leukocytes (particularly neutrophils and tissue-resident antigen-presenting populations, TRAP) and plasma proteins also play a role in shaping platelet activation dynamics and, ultimately, therapeutic performance [20,21].

## 3. Mechanisms and Drivers of Senescence

Building upon the established structural and molecular characteristics of senescent platelets, it becomes essential to understand the mechanisms that drive this process and the clinical environments in which it is accelerated. As previously introduced in Section 2, platelet aging is not exclusively limited to a random or linear decline in the chronological sense; it is, rather, a complex, stress-induced phenomenon shaped by cumulative exposure to biochemical insults and systemic imbalances. Among these, oxidative stress stands out as a primary mediator of senescence, exerting its effects through continuous reactive oxygen species exposure and the gradual breakdown of intracellular antioxidant defenses.

Hydrogen peroxide, in particular, which is generated via NADPH oxidase activity, plays a central role in the oxidative cascade that undermines platelet stability [22]. Over time, this promotes the oxidation of membrane lipids, cytoskeletal proteins, and mitochondrial components [23], all of which are critical for maintaining platelet activation capacity and granule release. As glutathione reserves are depleted and protein carbonylation accumulates, platelets become hyperreactive at rest and poorly responsive when physiologically activated [18]. Mitochondrial dysfunction further compounds these effects by impairing calcium handling and ATP generation, pushing senescent platelets into a low-energy, pro-degradation state [18,24]. Key structural and functional differences between young, competent platelets and their senescent counterparts are summarized in Table 1. It is important to distinguish senescence from activation, as the two processes are often conflated. Platelet activation is a rapid and typically reversible response to physiological or mechanical stimuli, triggering degranulation and mediator release. Senescence, on the other hand, is an irreversible, time- or stress-dependent decline in function, marked by mitochondrial exhaustion, granule depletion, and membrane instability. While rough handling during PRP preparation can induce transient activation or damage, this does not constitute true senescence.

These internal changes are further amplified by external stressors. Chronic inflammatory environments, especially those present in aging individuals and patients with metabolic dysfunction (metabolic syndrome), create a systemic backdrop that accelerates platelet turnover and depletes the circulating pool of competent platelets [18,25,26]. Interleukin (IL)-6 and tumor necrosis factor alpha (TNF-α), among other cytokines, have been implicated in promoting megakaryocyte stress and altering the lifespan of newly formed platelets [27,28]. As immune cells themselves age and accumulate a senescence-associated secretory phenotype (SASP), a self-reinforcing loop of inflammatory and oxidative pressure emerges, compromising platelet integrity across the lifespan [29].

Metabolic diseases such as diabetes provide a clear example of this dysregulation. In diabetic individuals, persistent hyperglycemia increases platelet production in an attempt to compensate for heightened destruction and turnover [30]. However, these newly generated platelets enter circulation in an environment dominated by glycation, oxidative damage, and mitochondrial stress [31]. The result would be paradoxical; despite often containing elevated levels of growth factors, such as VEGF [32], PRP derived from diabetic patients may exhibit a reduced therapeutic efficacy, likely due to underlying platelet dysfunction, poor metabolic functionality, and increased pro-inflammatory potential [30,31,33]. This impaired therapeutic response is not restricted to diabetes alone, but reflects a broader pattern observed in patients with metabolic syndrome and its associated comorbidities. As described in recent frameworks such as SDIMMMER [34], metabolic dysfunction creates a chronic, low-grade inflammatory state often referred to as ‘meta-inflammation’. This state disrupts tissue homeostasis, weakens anabolic signaling, and impairs cellular physiology across multiple systems [34]. In this context, regenerative interventions such as PRP are deployed on a biologically unfavorable terrain, where even optimally processed preparations may underperform due to the hostile systemic environment [34,35].

Similarly, cardiovascular diseases (CVD) and autoimmune conditions exert comparable stress on platelet biology, either by directly consuming platelets through thrombotic and immune-mediated mechanisms or by generating sustained oxidative and inflammatory burdens [36,37]. In all these settings, the cumulative result is the expansion of a senescent platelet phenotype that is not only less effective in regeneration, but potentially detrimental when included in orthobiologic formulations.

Finally, physiological aging introduces another layer of complexity. With age, the bone marrow microenvironment undergoes structural and molecular deterioration that limits the output and quality of hematopoietic progenitors, including megakaryocytes [38]. This leads to a reduced capacity for platelet renewal and an increased baseline of circulating senescent platelets [39,40]. When combined with chronic comorbidities and unfavorable lifestyle factors, this results in a high-risk profile for producing PRP with low regenerative potential [41], even in patients who would otherwise be considered suitable candidates for treatment.

Understanding these intertwined drivers of platelet senescence underscores the need for a personalized approach in regenerative medicine. Rather than viewing platelet aging as an inevitable consequence of time, it should be approached as a modifiable biological response to systemic stress. Therapeutic strategies such as the ‘Preparing the Soil’ concept and the ‘SDIMMMER’ approach [34,35] which focus on mitigating oxidative stress, preserving mitochondrial function, and modulating inflammation, may be key to maintaining platelet integrity and enhancing the clinical efficacy of PRP-based therapies. Although promising as a conceptual guide for patient preparation, the SDIMMMER framework, which was previously published in the literature, remains a theoretical construct that warrants validation through randomized controlled trials. Its practical clinical efficacy must still be established via systematic and reproducible investigations.

Table 2 summarizes current and emerging interventions aimed at enhancing platelet quality across different stages of PRP preparation.

## 4. Clinical and Functional Implications in PRP Therapy

As previously outlined, the clinical success of PRP therapies hinges not only on platelet concentration or growth factor release, but fundamentally on the functional integrity of the platelets being delivered. While traditional protocols emphasize variables such as numerical thresholds and activation strategies [3], recent insights into platelet senescence reveal a much more nuanced landscape. Platelets of equal quantity may differ dramatically in their therapeutic potential depending on their age, receptor expression, granule content, and responsiveness to stimuli [42].

Senescent platelets, characterized by structural fragility and reduced α-granule density, release their contents prematurely and inefficiently [12,17]. Logically, this leads to a diminished delivery of bioactive molecules at the target site, undermining the regenerative intent of the therapy. Moreover, these aged platelets may paradoxically contribute to local inflammation, displaying increased oxidative stress, disorganized signaling, and impaired aggregation profiles [10,43]. Therefore, instead of coordinating tissue repair, they may interfere with the regenerative cascade and potentially exacerbate tissue damage.

The ratio of young to aged platelets within a given PRP sample may represent a subtle yet critical determinant of therapeutic efficacy. While routine practice tends to prioritize total platelet counts, this focus often overlooks the functional heterogeneity of the population. In some cases, a preparation with fewer, but younger and more functionally competent platelets may outperform one with a higher concentration of senescent, dysfunctional cells. This suggests that, in regenerative therapies, more is not always better. Sometimes, less is more. Clinical variability observed among patients treated with seemingly identical PRP protocols may, in part, reflect this often-invisible biological variable.

A key advancement in assessing platelet age involves the use of Human Leukocyte Antigen-I (HLA-I) expression. Young platelets exhibit high levels of HLA-I, which progressively decline with senescence [14,44]. Flow cytometry studies using anti-HLA-I antibodies have successfully discriminated platelet subpopulations according to biological age, confirming the superior reactivity and biosynthetic activity of younger subsets [14,44]. While not yet fully integrated into routine clinical workflows, in the future, this approach may provide valuable clinical insight into the untapped potential of incorporating functional markers into PRP quality control. However, its clinical application currently remains limited due to the lack of standardized protocols, scalability challenges, and the absence of validation in large-scale outcome-driven studies.

It is worth noting that platelet age is not bound to the calendar. In a world where lifestyle and chronic disease shape biology, even youth can be deceptive. A young patient under metabolic duress may harbor platelets no more regenerative than those of an elderly individual. Despite rigorous attempts to establish global standardization in PRP protocols, therapeutic yield ultimately mirrors the biological state of the donor. No matter how refined the technique, a compromised biological starting point yields a compromised therapeutic product. Sometimes, it is not the method that fails but, rather, the material, especially when derived from a phenotypically senescent patient.

These perspectives challenge the validity of using a “one-size-fits-all” approach to PRP therapy. Physicians must look beyond processing tubes and begin treating systems, because systems under stress cannot regenerate what they themselves struggle to preserve. Such considerations become particularly relevant when examining specific clinical applications of PRP. In tendon and ligament repair, for instance, where sustained growth factor release and mechanical resilience are critical [45], the presence of senescent platelets with reduced degranulation capacity may lead to suboptimal tissue remodeling and an incomplete resolution of inflammation. Similarly, in cartilage regeneration, such as in osteoarthritis treatment, aged platelets may contribute to persistent synovial inflammation rather than resolution, due to their dysregulated cytokine profiles and impaired immunomodulatory functions. In dermatologic and wound healing contexts, platelet quality directly influences angiogenesis and re-epithelialization, with younger platelets promoting more efficient vascular and epithelial responses [46]. In peripheral nerve regeneration, the neurotrophic potential of PRP has been linked to its ability to deliver specific growth factors such as the brain-derived neurotrophic factor (BDNF) and nerve growth factor (NGF) [47], which, alongside other molecules, are often diminished in senescent platelet secretomes. These examples underscore the importance of aligning PRP composition with the biological demands of each clinical scenario and suggest that platelet age profiling may play a future role in tailoring therapy to the target tissue.

## 5. Strategies for Optimization and Platelet Quality Enhancement

Given the impact of platelet age and functionality on regenerative outcomes, optimizing the biological quality of platelets before PRP preparation becomes a critical step in clinical practice. Rather than relying solely on post-collection processing or high-volume blood draws, a more effective approach may be found in pre-procedural conditioning of the patient, targeted nutritional support, and refined laboratory techniques that prioritize platelet integrity and selective activation.

Pre-procedural optimization begins with addressing modifiable factors that influence platelet biogenesis and senescence. Nutritional status plays a central role [48]. For instance, key micronutrients such as folate, iron, vitamin B12, vitamin C, and omega-3 fatty acids contribute directly to megakaryocyte maturation, platelet formation, and antioxidant defense [48,49].

In this context, two recent clinical frameworks have been proposed to guide the optimization of autologous (and, where applicable, allogeneic) biologics prior to harvesting, including those derived from first-degree relatives. The “Preparing the Soil” model [35,50], for example, advocates for targeted interventions to improve metabolic health, reduce systemic inflammation, and restore cellular homeostasis prior to orthobiologic procedures. Similarly, the “SDIMMMER” approach [34] outlines a structured strategy to support cellular function, modulate the predominant pro-inflammatory status, and enhance the overall regenerative environment. Both models converge on the idea that enhancing the patient’s internal physiology, particularly in individuals with age-related decline or chronic comorbidities, can significantly improve the biological quality and functional competence of harvested components, including platelets and even other key cellular effectors and molecular mediators. By prioritizing systemic conditioning prior to orthobiologic harvesting, these strategies may offer a practical means of improving both the quality of the harvested biologic and the innate regenerative capacity of the host tissue, thereby producing synergistic effects. A stepwise workflow linking systemic preparation, precision harvesting, in-process quality control, and post-procedure feedback is summarized in Figure 2.

In parallel, mitigating oxidative stress through accessible strategies such as adopting regular moderate exercise, abstaining from alcohol and smoking, and avoiding substances known to impair platelet function may further enhance the biological quality of the platelet pool available for harvesting. These concepts align with the rationale proposed in the ‘Preparing the Soil’ and ‘SDIMMMER’ frameworks [34,35,50].

Beyond systemic factors, procedural variables such as anticoagulant choice and activation method also exert a decisive influence on the functional quality of the final PRP product [8,51,52,53]. Recent evidence has shown that changes in pH, for instance, as well as other processing-related conditions including temperature, can significantly affect platelet function, including both their hemostatic and immunomodulatory capacities [54]. Therefore, logically, these alterations may, in turn, also impact the stability of growth factors and the integrity of platelet activation mechanisms [54]. Although further studies are needed to clarify the full extent of these effects, maintaining consistency and precision during processing remains essential for preserving the regenerative potential of PRP. In this context, even biologically competent donor material may be rendered suboptimal by inadequate handling. Ensuring precision at every step, from source to separation, is essential for producing consistently high-quality PRP.

In terms of activation, alternatives such as calcium chloride (CaCl_2_) and autologous thrombin have gained clinical traction [52]. CaCl_2_, in particular, induces controlled platelet activation by providing extracellular calcium, triggering growth factor release without compromising membrane integrity [55]. When combined with autologous thrombin, it can further enhance fibrin formation and growth factor bioavailability, particularly in applications such as wound healing or soft tissue repair [56].

An emerging frontier in platelet optimization is the use of supernatant-based preparations, such as ‘supernatant of activated PRP’ (saPRP), which have been shown to exhibit reduced platelet senescence and apoptosis, alongside enhanced anti-inflammatory profiles, higher protein concentrations, and greater proliferative potential compared with conventional PRP formulations [57]. This technique, centered on activated platelet releasates, offers an avenue to harness the functional content of younger, biologically active platelets while minimizing interference from senescent subsets. This reduced influence is likely due to the fact that senescent platelets exhibit diminished degranulation capacities and compromised alpha granule contents, contributing less to the secretome generated upon activation. As a result, the composition of saPRP tends to reflect the output of functionally competent platelet populations.

In parallel, given that the bone marrow is the site of platelet biogenesis [58], aspirated marrow products may logically contain younger, pre-circulatory platelets, particularly in individuals without hematopoietic impairment. Since platelet senescence is largely induced after release into circulation, bone marrow-resident platelets retain a more functional phenotype [10]. However, given that platelets are rapidly released into circulation [12,59], their relative scarcity within the marrow compartment, combined with the technical demands of aspirate collection [60], limits the feasibility of using bone marrow aspirate (BMA) as a “standalone source for platelet-rich products”. While BMA does contain platelets, their concentration is generally lower than that of PRP, which is specifically processed to maximize platelet yield. Nonetheless, when used in combination with PRP, BMA-derived products may augment the overall regenerative profile of the formulation. This synergistic effect likely stems from the cellular richness of the bone marrow, which includes progenitor cells, stromal components, and immunomodulatory mediators that contribute to a more comprehensive biological response [61,62].

While cellular sources may offer upstream access to younger platelets, a parallel and less invasive strategy for supporting platelet function lies in photobiomodulation (PBM). This approach presents promising, albeit still exploratory, mechanisms for preserving or even enhancing platelet performance [63,64]. In addition to restoring mitochondrial balance and mitigating oxidative burden [65,66], PBM has been shown to reversibly attenuate platelet reactivity and reduce hemolysis [67], suggesting a role in preserving platelet integrity and delaying functional decline. Although direct evidence in PRP contexts is limited, these early findings point to PBM as a potential adjunctive strategy for maintaining platelet viability and optimizing the therapeutic profile of regenerative interventions.

In parallel with optimization strategies, assessing platelet quality remains a technical challenge in regenerative practice. Several methodologies have been proposed to evaluate functional platelet status, particularly in research contexts. Flow cytometry allows the detection of platelet activation, such as through P-selectin (CD62P) membrane localization or activated GPIIb/IIIa expression, and apoptosis, such as via Annexin V binding [68,69]. Mitochondrial membrane potential (ΔΨm) assays using JC-1 or TMRE staining, ATP quantification, and ROS detection are frequently used to infer bioenergetic competence [70,71,72]. Functional aggregometry, though rarely applied clinically, can offer insights into responsiveness. Despite their potential value, these methods lack standardization and remain largely unavailable in routine clinical workflows, highlighting the need for more accessible and validated tools to monitor platelet quality at the point of care.

Taken together, these strategies underscore a broader clinical shift from indiscriminate volume-based approaches to precision-driven, patient-centered protocols. The focus moves from extracting as many platelets as possible to selectively cultivating and deploying those with the highest therapeutic value. Optimizing the host environment, improving collection protocols, and incorporating advanced activation and processing techniques can all contribute to producing a PRP formulation that is not only rich in platelets, but in regenerative potential.

## 6. Future Directions and Clinical Translation

As the understanding of platelet senescence deepens, the path forward in regenerative medicine must evolve beyond empirical protocols and simplistic dose escalation. The future of PRP therapies lies in not only the precise selection of the right patients and indications, but also in refining the very cellular composition of what is being delivered. Quality must take precedence over quantity. Despite compelling preclinical evidence, large-scale randomized controlled trials directly correlating platelet age with clinical outcomes in regenerative therapies are still lacking. This remains a critical research gap before widespread clinical implementation can be considered.

One of the most promising avenues for clinical advancement involves the implementation of functional platelet profiling. Previously discussed techniques such as flow cytometry using HLA-I expression can distinguish platelet subpopulations based on biological age. This creates the possibility of selectively enriching preparations with the most therapeutically competent platelets, moving PRP closer to a targeted biologic rather than a generic concentrate. Although not yet feasible at the point of care, these tools may offer a glimpse into a future of personalized orthobiologics, where platelet quality is no longer inferred but directly measured and optimized. Indeed, the creation of a clinically viable platelet quality index would be an impactful development. While current techniques such as CD62P expression, ΔΨm assays, Annexin V binding, ROS detection, and HLA-I profiling offer quantifiable insights into platelet functionality and senescence, they remain largely confined to experimental settings. Future work should focus on integrating these measurable parameters into a composite score or profiling algorithm that could assist clinicians in evaluating PRP potency prior to administration. Such a tool could become essential for patient selection, protocol standardization, and improved treatment reproducibility in regenerative practice.

Simultaneously, pre-procedural patient conditioning should be formally integrated into clinical protocols. Nutritional support, antioxidant priming, and lifestyle adjustments may significantly alter the regenerative capacity of PRP. Rather than treating these variables as secondary, they should be viewed as fundamental to achieving reproducible outcomes. Likewise, combining PRP with adjunctive technologies such as tissue scaffolds or senescence-targeting compounds opens new doors for synergistic therapies that address both tissue damage and the biological limitations of autologous products.

A critical shift must also occur in how PRP volume and composition are conceptualized. The widespread practice of collecting large volumes of peripheral blood (often involving dozens of syringes and significant processing time) under the assumption that more input guarantees better therapeutic output is fundamentally flawed. High-volume protocols frequently lead to excessive anticoagulant exposure, reduced plasma recovery, and the inclusion of a higher proportion of senescent or pre-activated platelets. These formulations may not only fail to bring significant improvements but may also actively diminish therapeutic efficacy due to receptor saturation, biphasic dose–response effects, and premature growth factor release. Beyond the biological drawbacks, such protocols are resource-intensive, adding unnecessary procedural complexity and cost. Clinical efficacy is not a function of volume but of cellular precision. It is time to abandon the illusion that quantity can compensate for poor quality.

As new technologies emerge, including senolytic strategies, rejuvenation compounds, and machine learning models for predicting platelet age, the regenerative medicine field must embrace a more mechanistically informed, biologically grounded approach. Standardization efforts should expand beyond cell counts and include functional metrics such as platelet responsiveness, mitochondrial integrity, and granule content. Regulatory frameworks and product classification systems must also adapt to this evolving landscape, ensuring that quality control aligns with therapeutic intention.

In the coming years, the most effective PRP-based treatments may not be those that extract the most blood, but those that extract the most value from each competent cell. The paradigm is shifting, and with it, the responsibility of clinicians and researchers to rethink what defines an effective regenerative intervention. Cellular senescence, as a whole, is no longer just a scientific curiosity; it is a clinical variable with direct consequences. Recognizing and addressing it will be essential for advancing the next generation of orthobiologic therapies.

The concepts approached here aim to stimulate a deeper understanding of platelet age as another critical determinant of PRP efficacy. Nonetheless, we acknowledge that many of the strategies discussed are still in exploratory stages. Their effective clinical translation will require rigorous validation, standardization, and cost-effectiveness assessments before broader implementation can be justified.

## 7. Conclusions

Platelet-rich plasma therapies have reached a critical inflection point. While decades of research have focused on refining PRP formulations through adjustments in volume, concentration, and activation, a key biological determinant has remained largely unaddressed: platelet senescence. The functional decline that accompanies platelet aging, driven by oxidative stress, inflammation, and mitochondrial deterioration, compromises the regenerative capacity of PRP in ways that platelet count alone cannot predict.

This review highlights the urgent need to move beyond quantitative dogma and embrace a quality-centered approach. The presence of senescent platelets within PRP formulations undermines therapeutic potential, contributes to clinical variability, and challenges the notion that higher volumes yield better outcomes. Functional profiling of platelets, patient optimization protocols, and advanced processing strategies represent the next frontier in PRP science.

If platelet counts can be routinely measured to assess patient health status (and regenerative potential), why does platelet age still fall short of clinical consideration? By recognizing platelet age as a clinically relevant variable, practitioners can begin to tailor therapies that are not only autologous, but biologically precise. The future of PRP lies not in extracting more, but in extracting better. Physicians must remember that platelets may be abundant, but not all are equal. Some are like seasoned messengers still ready for action, while others, aged and damaged, have already lost the message they were meant to deliver.

## Figures and Tables

**Figure 1 cells-14-01206-f001:**
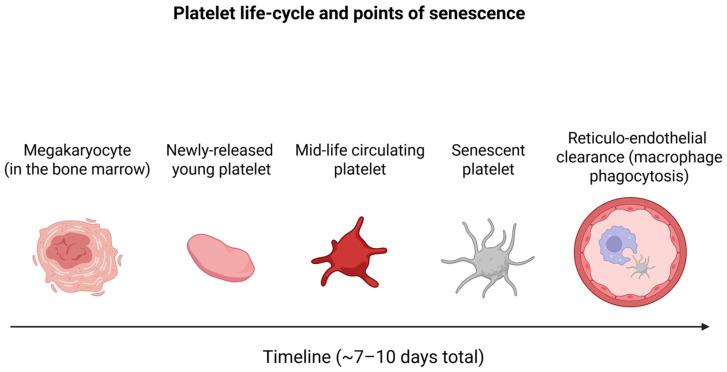
Schematic of platelet life-cycle stages from bone-marrow release to reticulo-endothelial clearance.

**Figure 2 cells-14-01206-f002:**
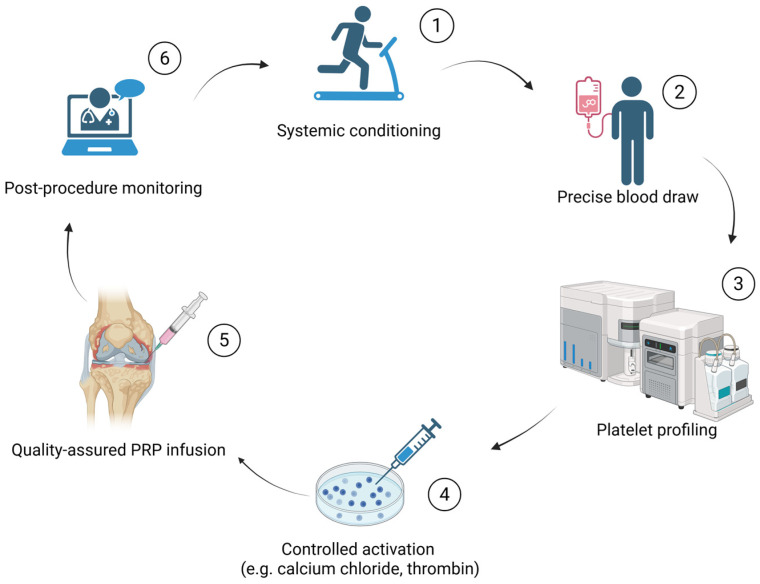
Stepwise PRP optimization workflow.

**Table 1 cells-14-01206-t001:** Key Features of Functional vs. Senescent Platelets.

Parameter	Functional Platelets	Senescent Platelets
Granule Content	Intact α and dense granules	Depleted granules
Mitochondrial Function	Efficient ATP production	Dysfunctional, ROS accumulation
Responsiveness	High reactivity to stimuli	Reduced responsiveness
Surface Markers	Normal CD62P, HLA-I	Increased phosphatidylserine, HLA-I loss
Cytokine Profile	Balanced, context-specific	Pro-inflammatory, dysregulated
Lifespan	~7–10 days	Shortened due to stress
Regenerative Potential	Supports repair	May inhibit or delay healing

**Table 2 cells-14-01206-t002:** Clinical and Technical Strategies to Optimize PRP Quality.

Strategy	Target	Mechanism
Nutritional Optimization	Systemic biology	Reduces oxidative stress and systemic inflammation
Antioxidant Supplementation	ROS balance	Protects cells
Moderate Exercise	Mitochondrial resilience	Improves cellular bioenergetics and metabolism
Minimizing Blood Draw Volume	Yield and quality	Reduces anticoagulant load
Precision in Processing	Cellular preservation	Maintains pH, temperature, stability
Photobiomodulation	Cellular function	Modulates mitochondrial function

## Data Availability

No new data were created or analyzed in this study.

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
