# Peer review of "Not All Platelets Are Created Equal: A Review on Platelet Aging and Functional Quality in Regenerative Medicine"

_cells, 2025, doi:10.3390/cells14151206_

Round 1

Reviewer 1 Report

Comments and Suggestions for Authors

The authors discussed the efficacy of PRP therapy based on the platelet quality. This study is of interest and is indispensable for improving the PRP therapy. To disseminate this concept more widely in the field of regenerative medicine, this reviewer encourages the authors (as well as others) to share their views and expand this discussion in various ways.

However, this concept is not new in itself and has long been shared by many clinicians and researchers without substantial evidence, particularly those concerned with the lower predictability of PRP therapy. Thus, this reviewer wonders whether it is now time to release a review article, given the limited scientific evidence. This article may be acceptable as “prospect”, “opinion”, or “proposal”, but not “narrative review.”

The primary concerns of this reviewer are listed below.

  • Are individual differences in platelet responsiveness to activation stimuli predominantly dependent on the platelet quality?
  • This study lacks the information regarding methodologies for evaluating platelet quality, including their strengths and weaknesses.
  • The biological characteristics of senescent platelets are acceptable. However, it is questionable whether these characteristics can be used as markers of senescent platelets. Which criteria, the molecular hallmarks mentioned by the authors, can identify senescent platelets?
  • This concept lacks a quantitative point for future clinical applications, especially in the evaluation of platelet quality.
  • The authors suggested clinical and technical strategies to optimize the PRP quality (Table 2). The reviewer agrees with their idea of a more reliable PRP therapy. However, there is no doubt that these strategies first improve systemic conditions to enhance the innate regenerative capacity, regardless of the application of PRP. This may be more important than the changes in platelet quality.
  • The authors have referred to the supernatant of the activated PRP on page 8. However, it isn't easy to understand why senescent platelets rarely influence this preparation. Additional explanations will help readers’ understanding.
  • Are activated platelets, particularly those undergoing reversible mild activation, considered aging platelets? Please explain how activated and senescent platelets can be distinguished. Mechanical stimuli can induce mild activation during rough preparation of PRP and may cause platelet “senescence.”

We recommend that the authors first investigate and validate potential biomarkers for identifying senescent platelets or quantitatively evaluating platelet quality.

Author Response

Reviewer 1

Reviewer wrote:

“Comments and Suggestions for Authors

The authors discussed the efficacy of PRP therapy based on the platelet quality. This study is of interest and is indispensable for improving the PRP therapy. To disseminate this concept more widely in the field of regenerative medicine, this reviewer encourages the authors (as well as others) to share their views and expand this discussion in various ways.

However, this concept is not new in itself and has long been shared by many clinicians and researchers without substantial evidence, particularly those concerned with the lower predictability of PRP therapy. Thus, this reviewer wonders whether it is now time to release a review article, given the limited scientific evidence. This article may be acceptable as “prospect”, “opinion”, or “proposal”, but not “narrative review.”

The primary concerns of this reviewer are listed below.

  • Are individual differences in platelet responsiveness to activation stimuli predominantly dependent on the platelet quality?
  • This study lacks the information regarding methodologies for evaluating platelet quality, including their strengths and weaknesses.
  • The biological characteristics of senescent platelets are acceptable. However, it is questionable whether these characteristics can be used as markers of senescent platelets. Which criteria, the molecular hallmarks mentioned by the authors, can identify senescent platelets?
  • This concept lacks a quantitative point for future clinical applications, especially in the evaluation of platelet quality.
  • The authors suggested clinical and technical strategies to optimize the PRP quality (Table 2). The reviewer agrees with their idea of a more reliable PRP therapy. However, there is no doubt that these strategies first improve systemic conditions to enhance the innate regenerative capacity, regardless of the application of PRP. This may be more important than the changes in platelet quality.
  • The authors have referred to the supernatant of the activated PRP on page 8. However, it isn't easy to understand why senescent platelets rarely influence this preparation. Additional explanations will help readers’ understanding.
  • Are activated platelets, particularly those undergoing reversible mild activation, considered aging platelets? Please explain how activated and senescent platelets can be distinguished. Mechanical stimuli can induce mild activation during rough preparation of PRP and may cause platelet “senescence.”

We recommend that the authors first investigate and validate potential biomarkers for identifying senescent platelets or quantitatively evaluating platelet quality.”

Reply:

We sincerely thank the reviewer for the encouraging comments and thoughtful critique. We greatly appreciate the opportunity to clarify our rationale and address the points raised.

Regarding the general comment on the originality and format, while we agree that the broader concept of “platelet quality affecting PRP outcomes” has been informally shared by many clinicians, the specific role of platelet senescence has not been widely addressed in the literature. In fact, our own motivation for writing this article came from a lack of structured discussions on this exact topic. Most reviews on PRP either focus on clinical protocols or growth factor quantification, with very few exploring senescence-related molecular features, mitochondrial impairment, or donor aging as contributors to variability.

We believe this provides justification for the review’s narrative nature, even though we understand and respect the reviewer’s position that the article might also qualify as a perspective or conceptual proposal. We are entirely open to reclassifying the manuscript if the editorial team prefers it to fall under a different category.

We address the specific points below:

  1. Platelet responsiveness and platelet quality:
    We agree that platelet responsiveness is multifactorial. We’ve clarified in Section 2 that platelet quality plays a central role but is not the only determinant.
  2. Lack of methodology for assessing quality:
    We added a brief overview of current assessment tools (flow cytometry markers, ΔΨm, ROS, PS exposure, etc.) in Section 5, along with their limitations.
  3. Criteria for senescent platelets:
    We thank the reviewer for raising this important point. The molecular criteria currently available for identifying senescent platelets are indeed limited, but a key emerging biomarker is Human Leukocyte Antigen-I (HLA-I), as discussed in Section 4 of our manuscript. We describe how high levels of HLA-I expression correlate with younger platelet subpopulations, and how flow cytometry using anti-HLA-I antibodies has been successfully used to discriminate platelet age. While not yet standardized for routine clinical use, this marker offers a valuable basis for further exploration and is aligned with the review’s objective of identifying promising avenues for future research.
  4. Lack of quantitative thresholds:
    We thank the reviewer for highlighting the importance of establishing quantitative parameters for platelet quality. As correctly noted, current methods such as flow cytometry for CD62P expression, ΔΨm assays, and ROS detection offer valuable insights, but remain confined to research settings due to standardization and scalability barriers. We have now clarified in the revised manuscript that while no unified clinical score exists yet, these quantifiable metrics form the basis for a future platelet quality index that could support patient selection and procedural planning.
  5. Systemic optimization vs. PRP composition:
    We appreciate the reviewer’s thoughtful comment. We agree that systemic optimization strategies such as nutritional support and inflammation control not only improve platelet biogenesis and quality but also promote broader regenerative capacity, regardless of PRP application. Our intention was to frame these interventions as synergistic rather than PRP-specific. As emphasized in the discussion of the “Preparing the Soil” and “SDIMMMER” models, a well-conditioned host environment enhances both the autologous biologic and the tissue’s ability to respond. We have clarified this dual role in Section 3.
  6. Senescent platelets in PRP supernatant:
    We appreciate the reviewer’s request for clarification. The reduced influence of senescent platelets on saPRP is based on their diminished capacity to degranulate and release bioactive molecules upon activation. Since the supernatant is composed primarily of released factors following activation (e.g., growth factors, cytokines), the contribution of senescent platelets (whose alpha and dense granule content is often depleted or dysfunctional) is inherently lower compared to younger, functionally active platelets. We have expanded the explanation in Section 4 accordingly.
  7. Activation vs. senescence:
    We appreciate the reviewer’s insightful question regarding the distinction between platelet activation and senescence, particularly in the context of PRP preparation. While these processes may share overlapping triggers or surface markers, they are biologically distinct. Platelet activation is a rapid, typically reversible response that enables degranulation and signaling in response to physiological or mechanical stimuli. In contrast, senescence is an irreversible, stress- or time-driven decline in function, characterized by mitochondrial dysfunction, granule depletion, and membrane instability. To clarify this distinction, we have added a dedicated paragraph in Section 3 (Mechanisms and Drivers of Senescence) right before the introduction of Table 1.

Again, we thank the reviewer for their constructive input. We hope these clarifications help align the intent of the manuscript with the reviewer's expectations, and we remain open to any reclassification that the editorial board deems more appropriate.

Reviewer 2 Report

Comments and Suggestions for Authors

The review by Costa et al is focused on the quality of platelet preparations looked upon through the effects of platelet age-related properties (dependent not only on the age per se) on the effectiveness of these preparations in clinical practice. The mechanisms underlying these age-related changes and the approaches to counteract the changes detrimental to the practical use of platelet preparations are thoroughly discussed. The review is of interest to many researchers working in the platelet area. 

To further improve the review, I would suggest to more clearly outline various clinical applications of platelet preparations and explicitly indicate how the platelet age-related qualities of these preparations affect individual uses.

Author Response

We thank the reviewer for the positive evaluation and constructive suggestion. We agree that the clinical applicability of PRP formulations may be differentially influenced by platelet quality depending on the target tissue and therapeutic goal. In response, we have expanded Section 4 (Clinical and Functional Implications in PRP Therapy) to more clearly outline various clinical uses of PRP, such as in musculoskeletal, dermatologic, and neurologic indications. We have also added commentary on how platelet age and senescence may differentially impact these applications, especially regarding the importance of sustained growth factor delivery, anti-inflammatory modulation, and platelet-tissue interaction profiles.

Round 2

Reviewer 1 Report

Comments and Suggestions for Authors

This reviewer appriciated that you have appropriately addressed the comments and made sufficient revisions. This reviewer is looking forward to your original research on this topic.

Author Response

We sincerely thank the reviewer for the positive feedback and for acknowledging our revisions. We greatly appreciate the constructive comments that helped us improve the clarity and depth of our manuscript. Your encouraging words regarding our future original research are highly motivating, and we are grateful for your support.